# Molecular Characterizations of the *er1* Alleles Conferring Resistance to *Erysiphe pisi* in Three Chinese Pea (*Pisum sativum* L.) Landraces

**DOI:** 10.3390/ijms231912016

**Published:** 2022-10-10

**Authors:** Suli Sun, Dong Deng, Wenqi Wu, Yuhua He, Gaoling Luo, Chengzhang Du, Canxing Duan, Zhendong Zhu

**Affiliations:** 1Institute of Crop Sciences, Chinese Academy of Agricultural Sciences, Beijing 100081, China; 2Yunnan Academy of Agricultural Sciences, Kunming 650205, China; 3Rice Research Institute, Guangxi Academy of Agricultural Sciences, Nanning 530007, China; 4Institute of Specialty Crop, Chongqing Academy of Agricultural Sciences, Chongqing 402160, China

**Keywords:** *Erysiphe pisi*, *er1*-13, *er1*-14, KASPar marker, pea

## Abstract

Powdery mildew caused by *Erysiphe pisi* DC. is a major disease affecting pea worldwide. This study aimed to confirm the resistance genes contained in three powdery mildew-resistant Chinese pea landraces (Suoshadabaiwan, Dabaiwandou, and Guiwan 1) and to develop the functional markers of the novel resistance genes. The resistance genes were identified by genetic mapping and *PsMLO1* gene sequence identification. To confirm the inheritance of powdery mildew resistance in the three Landraces, the susceptible cultivars Bawan 6, Longwan 1, and Chengwan 8 were crossed with Suoshadabaiwan, Dabaiwandou, and Guiwan 1 to produce F_1_, F_2_, and F_2:3_ populations, respectively. All F_1_ plants were susceptible to *E**. pisi*, and phenotypic segregation patterns in all the F_2_ and F_2:3_ populations fit the 3:1 (susceptible: resistant) and 1:2:1 (susceptible homozygotes: heterozygotes: resistant homozygotes) ratios, respectively, indicating powdery mildew resistance in the three Landraces were controlled by a single recessive gene, respectively. The analysis of *er1*-linked markers and genetic mapping in the F_2_ populations suggested that the recessive resistance genes in three landraces could be *er1* alleles. The cDNA sequences of 10 homologous *PsMLO1* cDNA clones from the contrasting parents were obtained. A known *er1* allele, *er1*-4, was identified in Suoshadabaiwan. Two novel *er1* alleles were identified in Dabaiwandou and Guiwan 1, which were designated as *er1*-13 and *er1*-14, respectively. Both novel alleles were characterized with a 1-bp deletion (T) in positions 32 (exon 1) and 277 (exon 3), respectively, which caused a frame-shift mutation to result in premature termination of translation of *PsMLO1* protein. The co-dominant functional markers specific for *er1*-13 and *er1*-14, KASPar-*er1*-13, and KASPar-*er1*-14 were developed and effectively validated in populations and pea germplasms. Here, two novel *er1* alleles were characterized and their functional markers were validated. These results provide powerful tools for marker-assisted selection in pea breeding.

## 1. Introduction

Pea (*Pisum sativum* L.) is an important and widely distributed cool season legume crop, which frequently suffers from abiotic and biotic stresses during the whole growth season [1,2]. Among the biotic factors, the disease is the main cause affecting pea production [2]. Powdery mildew caused by *Erysiphe pisi* DC. is a major constraint for pea yield and quality worldwide [3]. *E. pisi* infections of peas can lead to yield losses of up to 80% in regions that are suitable for disease infection [3]. To date, the use of resistant cultivars carrying the *E. pisi*-resistant gene *er1* has been considered to be the most effective and environmentally friendly way to control this disease [4,5].

*E. pisi*-resistance in pea has been proved to be controlled by three different genes in different germplasms, including two single recessive genes (*er1* and *er2*) and one dominant gene (*Er3*) [6,7,8,9]. The *er1*, *er2*, and *Er3* genes have been mapped on different linkage groups of peas using linked markers [10,11,12,13,14,15,16,17,18]. The two recessive genes *er1* and *er2* were mapped to pea linkage groups (LGs) VI and III, respectively [10,19]. The dominant gene *Er3* isolated from wild pea (*P**isum*
*fulvum*) was located to pea LG IV recently [20].

To date, the recessive gene *er1* is the most widely used gene in pea production due to *er1* conferring high resistance or immunity to *E. pisi* in most pea germplasms [21]. In contrast, resistance conferred by *er2* is unstable and easily affected by leaf development stage and plant location [7,21,22,23]. *er2* is only found in a few pea germplasms [21]. *Er3* was known from wild pea (*P. fulvum*), and there have not been any extensive studies conducted to date [8,24].

Gene *er1* inhibits the incursion of *E. pisi* into pea epidermal cells, which confers stable and durable resistance to *E. pisi* [23]. Recent studies have shown that the *er1*-resistant phenotype is caused by loss-of-function mutations in the pea *MLO* (Mildew Resistance Locus O) homolog (*PsMLO1*). The *MLO* gene family has been identified in both dicots (e.g., *Arabidopsis thaliana*, and tomato—*Solanum lycopersicum*) and monocots (e.g., barley—*Hordeum vulgare*) [9,25,26,27,28,29].

To date, total of 12 *er1* alleles have been identified conferring resistance to *E. pisi* by natural mutation or obtained by mutagenesis in pea germplasms: *er1*-1 (also known as *er1mut1*) [9,13,16,30,31], *er1*-2 [9,15,16], *er1*-3 [9], *er1*-4 [9], *er1*-5 [28], *er1*-6 [18], *er1*-7 [17], *er1*-8, *er1*-9 [32], *er1*-10 (also known as *er1mut2*) [13,30,33], and *er1*-11 [33,34]; *er1*-12 was more recently identified in pea germplasm JI2019 from India [35]. Each *er1* allele corresponds to a different *PsMLO1* mutation site and pattern. Among the 12 *er1* alleles identified, only *er1*-1 and *er1*-2 are commonly applied in pea breeding programs [9,28]. Previous studies revealed that the functional markers of the known *er1* alleles have been developed and applied for the rapid selection of pea germplasms resistant to *E. pisi* in pea breeding [15,17,18,28,33,34,36].

*E. pisi* severely affects the yield and quality of pea crops in China [2]. The disease infects up to 100% of pea plants in some regions of planting susceptible cultivars. In our previous studies, we have focused on the identification of pea germplasms resistant to *E. pisi* [31,37]. A novel *er1* allele *er1*-6 had been identified in a Chinese pea germplasm [17] and new alleles *er1*-7, *er1*-8, and *er1*-9 were identified in pea germplasms from India, Afghanistan, and Australia, respectively [17,32]. *er1*-6 was also identified in some pea landraces from Yunnan Province of China [18]. Thus, a natural mutation of the *er1* gene conferring *E. pisi*-resistance has been observed in some Chinese pea landraces, which provides rich resistant sources that can be used to improve the *E. pisi* resistance of Chinese pea cultivars. The allelic diversity of this locus in the cultivated pea has been well characterized; however, relatively few studies have investigated and characterized the *E. pisi*-resistant gene in Chinese pea landraces. Thus, this study aimed to identify and characterize the *E. pisi*-resistant gene in three *E. pisi*-resistant Chinese pea landraces by genetic mapping and homologous *PsMLO1* gene sequence cloning. Additionally, any novel *er1* alleles were performed to develop their functional markers to improve marker-assisted selection in *E. pisi*-resistant pea breeding programs.

## 2. Results

### 2.1. Phenotypic Evaluation and Inheritence Analysis for Resistance

Six parental cultivars and contrasting controls were evaluated for their resistance to the *E. pisi* isolate EPYN. At 10 days post-inoculation, the *E. pisi* disease severity of the susceptible control was rated as score 4, indicating susceptibility to *E. pisi*. As expected, the three resistant pea parents, Suoshadabaiwan, Dabaiwandou, and Guiwan 1, and resistant control (Xucai 1) were immune to *E. pisi* infection (disease severity 0), while the three susceptible parents (Bawan 6, Longwan 1, and Chengwan 8) were susceptible to *E. pisi* (disease severity 4) (Figure 1). The segregation patterns of *E. pisi* resistance in the F_1_, F_2_, and F_2:3_ populations derived from the crosses Bawan 6 × Suoshadabaiwan, Longwan 1 × Dabaiwandou, and Chengwan 8 × Guiwan 1 are presented in Table 1.

Five F_1_ plants produced from the cross Bawan 6 × Suoshadabaiwan were susceptible to *E. pisi* (Table 1). One of the five plants generated 102 F_2_ and F_2:3_ offspring through self-pollination. Of these 102 F_2_ plants, 26 were resistant (R) to *E. pisi,* and 76 were susceptible (S) to *E. pisi*. This indicates that the segregation ratio (resistance: susceptibility) in the F_2_ population was 1:3 (χ^2^ = 0.02; *p* = 0.88), indicating the inheritance of a single recessive gene. Moreover, a segregation ratio of 26 (homozygous resistant):51 (segregating):25 (homozygous susceptible) in the F_2:3_ population fitted well with the genetic model of 1:2:1 ratio (χ^2^ = 0.03, *p* = 0.99) (Table 1), confirming that the *E. pisi* resistance in Suoshadabaiwan was controlled by a single recessive gene.

The cross of Longwan 1 × Dabaiwandou generated six F_1_ plants, which showed *E. pisi*-susceptibility (Table 1). One of six F_1_ plants generated 121 F_2_ offspring. Of 121, 29 were resistant, and 92 of 121 were susceptible to *E. pisi*. The segregation ratio in the F_2_ population of resistance to susceptibility fitted a genetic model ratio of 1:3 (χ^2^ = 0.07; *p* = 0.79), also indicating the inheritance of a single recessive gene. Moreover, a segregation ratio of 29 (homozygous resistant):56 (segregating):36 (homozygous susceptible) in the F_2:3_ population (121 families) fitted well with the genetic model of 1:2:1 ratio (χ^2^ = 1.41; *p* = 0.49), indicating that *E. pisi* resistance in Dabaiwandou was also controlled by a single recessive gene (Table 1).

The cross of Chengwan 8 × Guiwan 1 generated eight F_1_ plants which showed *E. pisi*-susceptibility (Table 1). One of eight F_1_ plants generated 131 F_2_ offspring. Of 131, 36 were resistant, and 95 of 131 were susceptible to *E. pisi*. The segregation ratio in the F_2_ population of resistance to susceptibility fitted a genetic model ratio of 1:3 (χ^2^ = 0.43; *p* = 0.51), also indicating the recessive inheritance of a single gene. Moreover, a segregation ratio of 36 (homozygous resistant):61 (segregating):34 (homozygous susceptible) in the F_2:3_ population (131 families) fitted well with the genetic model of 1:2:1 ratio (χ^2^ = 0.67; *p* = 0.71), indicating that *E. pisi* resistance in Guiwan 1 was also controlled by a single recessive gene (Table 1).

### 2.2. Mapping of Resistance Genes

Of the molecular markers tested, six (c5DNAmet, AD160, AC74, AD51, AD59, and AD60) were polymorphic between contrasting parents Bawan 6 and Suoshadabaiwan, and three (c5DNAmet, AA220, and AD51) were polymorphic between Longwan 1 and Dabaiwandou, Unfortunately, no polymorphic marker appeared between Longwan 1 and Dabaiwandou among the above markers tested. Thus, the additional eight SSR markers (16410, 28516, 26140, 23309, 29872, 26514, 23949, and 22903) developed recently were used to test the polymorphism between Longwan 1 and Dabaiwandou [38]. Two (26514 and 22903) were polymorphic between the contrasting parents, Longwan 1 and Dabaiwandou. All polymorphic markers between the parents were likely linked to the *E. pisi* resistance gene, respectively. Thus, the six, three, and the two parental polymorphic markers were used to confirm the genotypes of each F_2_ plant derived from Bawan 6 × Suoshadabaiwan, Longwan 1 × Dabaiwandou, and Chengwan 8 × Guiwan 1, respectively. This genetic linkage analysis suggested that six markers (c5DNAmet, AD160, AC74, AD51, AD59, and AD60), three markers (c5DNAmet, AA220, and AD51), and two markers (26514 and 22903) were linked to the resistance gene *er1* in Suoshadabaiwan, Dabaiwandou, and Guiwan 1, respectively (Figure 2). Our results also indicated that the resistance genes in the three resistant cultivars were located in the *er1* region. In Suoshadabaiwan, the linkage map indicated that the markers (AD59 and AD60) were mapped on both sides of the target gene with 3.4 cM and 8.3 cM genetic distances, respectively (Figure 2A). In Dabaiwandou, two other markers (c5DNAmet and AA220) were located on both sides of the target gene with 2.6 cM and 11.6 cM genetic distances, respectively (Figure 2B). In Guiwan 1, two markers (26514 and 22903) were located on both sides of the target gene with 12.8 cM and 19.3 cM genetic distances, respectively (Figure 2C). Our linkage and genetic map analyses confirmed that an *er1* allele controlled *E. pisi* resistance in Suoshadabaiwan, Dabaiwandou, and Guiwan 1, respectively (Figure 2).

### 2.3. PsMLO1 Sequence Analysis

The *PsMLO1* cDNA sequences of Bawan 6, Longwan 1, Chengwan 8, and the susceptible parents, were consistent with that of the wild-type *PsMLO1* cDNA.

In landrace Suoshadabaiwan, a 1-bp deletion (A) was identified in a previously reported position 91 in exon 1 of the *PsMLO1* cDNA sequence. This result is consistent with the mutation in the *er1* gene carried by germplasm YI (JI1591), named *er1*-4. In landrace Dabaiwandou, a novel mutation pattern was found in the Dabaiwandou cDNA fragment homologous to *PsMLO1*: a 1-bp deletion (T) corresponding to positions 32 in exon 1 (the first exon) of the *PsMLO1* cDNA sequence. This deletion caused a substitution of the amino acid leucine with tryptophan at position 11 of the PsMLO1 protein sequence (Figure 3A). This change caused the early termination of protein translation, probably also resulting in a functional change of PsMLO1 (Figure 3A). In Guiwan 1, a 1-bp deletion (T) was also identified in a previously unreported position 277 in exon 3 of the *PsMLO1* cDNA sequence. This deletion caused a substitution of the amino acid tryptophan with glycine at position 93 of the PsMLO1 protein sequence (Figure 3B). This change caused the early termination of protein translation, probably also resulting in a functional change of PsMLO1 (Figure 3B). The two natural mutations differed from all known *er1* alleles, indicating that the *E. pisi* resistance of Dabaiwandou and Guiwan 1 was controlled by the novel alleles of *er1*. These novel alleles were designated *er1*-13 and *er1*-14, respectively, following the accepted nomenclature [9,17,18,32,33,35,36]. Thus, a known and two novel *er1* alleles were discovered in the three resistant cultivars, Suoshadabaiwan (from Chongqing), Dabaiwandou (from Yunnan), and Guiwan 1 (from Guangxi), respectively.

### 2.4. Development of Functional Markers for er1-13 and er1-14

The KASPar markers flanking the 1-bp (T) deletion sequences from Dabaiwandou and Guiwan 1 were designed as functional markers specific for KASPar-*er1*-13 and KASPar-*er1*-14, respectively.

As expected, KASPar-*er1*-13 and KASPar-*er1*-14 successfully distinguished the contrasting parents (Longwan 1 and Dabaiwandou, Chengwan 8 and Guiwan 1) into two different clusters corresponding to the FAM-labeled and HEX-labeled groups in the Kompetitive allele-specific PCR (KASPar) assay, respectively. When KASPar-*er1*-13 and KASPar-*er1*-14 were used to analyze the 121 and 131 F_2_ progeny derived from Longwan 1 × Dabaiwandou and Chengwan 8 × Guiwan 1, the KASPar markers clearly separated the F_2_ progeny into three clusters corresponding to three genotypes: homozygous resistant, homozygous susceptible, and heterozygous (Figure 4). In the F_2_ population derived from Longwan 1 × Dabaiwandou, 29 plants were identified as homozygous resistant, 56 were heterozygous, and 36 were homozygous susceptible. In the F_2_ population derived from Chengwan 8 × Guiwan 1, 36 plants were homozygous resistant, 61 were heterozygous, and 34 were homozygous susceptible. These results were completely consistent with the phenotypes of both F_2:3_ populations, suggesting that KASPar-*er1*-13 and KASPar-*er1*-14 co-segregated with *er1*-13 and *er1*-14, respectively. A chi-squared (χ^2^) test showed that both segregation ratios of KASPar-*er1*-13 and KASPar-*er1*-14 in respective F_2_ populations fit 1:2:1 (KASPar-*er1*-13: χ^2^ = 1.41, *p* = 0.49; KASPar-*er1*-14: χ^2^ = 0.67; *p* = 0.71), indicating co-dominant markers.

### 2.5. Validation and Application of Functional Markers

The 56 germplasms with the known resistance phenotype to *E. pisi* isolate EPYN that carrying the known *er1* alleles (*er1*-1, *er1*-2, *er1*-4, *er1*-6, *er1*-7, *er1*-8, and *er1*-9) were selected to genotyping by KASPar-*er1*-13 and KASPar-*er1*-13 (Appendix A). It included 49 that were phenotypically immune to *E. pisi* and contained known *er1* alleles; seven were resistant. The three resistant and three susceptible parents were also tested at the same time (Appendix A).

When the 56 germplasms were genotyped with KASPar-*er1*-13, two distinct clusters were recovered, with one gene (*er1*-13) corresponding to Dabaiwandou and the other (non-*er1*-13) to the other germplasms, respectively. Similarly, when the germplasms were genotyped with KASPar-*er1*-14, two distinct clusters were recovered, corresponding to Guiwan 1 and all of the other germplasms, respectively (Figure 4; Appendix A). Thus, markers KASPar-*er1*-13 and KASPar-*er1*-14 effectively identified pea germplasms carrying the *er1*-13 and *er1*-14 alleles, respectively. Our results also showed that KASPar-*er1*-13 and KASPar-*er1*-14 could distinguish the know *er1* alleles and *Er1* from *er1*-13 or *er1*-14, respectively.

## 3. Discussion

Pea powdery mildew caused by *E. pisi* DC. is an important disease and reduces considerable yield in pea production worldwide. The deployment of resistant cultivars containing the *er1* gene is the most effective way to control this disease The *E. pisi* resistance gene *er1* is recessive in pea cultivars, which is the most widely deployed gene for controlling powdery mildew worldwide.

To date, there were 12 *er1* alleles identified in resistant pea germplasms. Among the 12 known *er1* alleles, *er1*-1 and *er1*-2 are most commonly used in pea breeding programs because they confer stable resistance to *E. pisi* [9,16,28,39,40]. Previously, *er1*-1 has been identified in four *E. pisi*-resistant pea cultivars (JI1559, Tara, and Cooper from Canada; and Yunwan 8 from China), while *er1*-2 has been identified in seven *E. pisi*-resistant pea cultivars (Stratagem, Franklin, Dorian, Nadir, X9002, Xucai 1, and G0005576) [9,15,16,18,28]. Recently, the *er1* gene for *E. pisi* resistance was confirmed in 53 pea germplasms from a worldwide collection [32]. Here, more *E. pisi*-resistant germplasms carrying the *er1*-1, *er1*-2, *er1*-6, and *er1*-7 alleles were identified. To date, a dominant gene *Er3* had been found in wild pea (*P**isum*
*fulvum*) and mapped to pea LG IV [20]. It is possible that a rich diversity of *E. pisi*-resistant genes were contained in wild pea. Thus, searching for novel *E. pisi*-resistant genes from wild pea germplasms should be a good strategy [8].

To date, more than 40 *MLO* mutant alleles have been described in the monocotyledonous plant barley [41], and *PsMLO1* allelic diversity has been widely studied in pea [9,13,16,17,18,28,30,31,32,33,34,35,36,40]. Wild-type *PsMLO1* of pea consists of 15 exons and 14 introns (NCBI accession number: KC466597). To date, a total of 12 *er1* alleles associated with the *er1*-resistance phenotype have been identified and 11 of 12 *er1* alleles *PsMLO1* mutations were caused by alterations of the coding sequence. There was 1, 1, 1, 1, 3, 1, and 2 allele mutations occurred in exons 1, 5, 6, 8, 10, 11, and 15 of wild-type *PsMLO1*, respectively. Eight alleles (*er1*-1/*er1mut1*, *er1*-3, *er1*-4, *er1*-5, *er1*-6, *er1*-9, *er1*-10/*er1mut2*, and *er1*-12) are the result of point mutations in the exons of wild-type *PsMLO1*. Four alleles result from single base substitutions in wild-type *PsMLO1* cDNA: in *er1*-1, a C→G at position 680 (exon 6); in *er1*-5, a G→A at position 570 (exon 5); in *er1*-6, a T→C at position 1121 (exon 11); and in *er1*-10, a G→A at position 939 (exon 10) [9,18,28,30]. Three alleles result from single base deletions in wild-type *PsMLO1* cDNA, including ΔG at position 862 (exon 8) in *er1*-3; ΔA at position 91 (exon 1) in *er1*-4; and ΔT at position 928 (exon 10) in *er1*-9 identified in this study [9]. Recently, *er1*-12 allele was found resulting from single base insertion (A) in front of the last exon (exon 15) in wild-type *PsMLO1* cDNA [35]. Two alleles result from small fragment deletions in wild-type *PsMLO1* cDNA, including a 10-bp deletion of positions 111–120 (exon 1) in *er1*-7 [17]; and a 3-bp deletion of positions 1339–1341 (exon 15) in *er1*-8 [32]. Only the *er1*-11 mutation is known to have resulted from an intron mutation in *PsMLO1* (a 2-bp insertion in intron 14) [33,34], and only *er1*-2 results from a large insert of unknown size in wild-type *PsMLO1* cDNA [9,15,18].

Several functional markers specific to the previously recognized *er1* alleles have already been developed to facilitate the marker-assisted breeding of pea cultivars resistant to *E. pisi* [9,15,17,18,30,32,33,34,36]. Pavan et al. [28] developed a functional cleaved amplified polymorphic sequence (CAPS) marker for *er1*-5, while Pavan et al. [36] developed functional markers for the five *er1* alleles, *er1*-1 through *er1*-5. Santo et al. [30] developed functional markers for *er1mut1* and *er1mut2*, and Wang et al. [15] developed a dominant marker for *er1*-2. Sudheesh et al. [34] developed a functional marker for *er1*-11, while Sun et al. [17,18] developed co-dominant functional markers for *er1*-6 and *er1*-7. Ma et al. [33] developed eight KASPar markers for eight known *er1* alleles, excluding *er1*-2, and renamed the *er1*-10 and *er1*-11. More recently, Sun et al. [32] identified the two novel *er1* alleles, *er1*-8 and *er1*-9, and developed KASPar-*er1*-8 and KASPar-*er1*-9. In this study, the developed markers, KASPar-*er1*-13 and KASPar-*er1*-14, could accurately and effectively identify pea germplasms carrying the *er1*-13 and *er1*-14 alleles and distinguish them from the know *er1* alleles or *Er1*, respectively.

This study discovered a known and two novel *er1* alleles, resulting from new mutations of wild-type *PsMLO1* cDNA: *er1*-13 and *er1*-14 was generated by a 1-bp deletion in exon 1 and 3, respectively. The co-dominant functional markers specific to *er1*-13 (KASPar-*er1*-13) and to *er1*-14 (KASPar-*er1*-14) were developed. These markers were validated in genetic populations and pea germplasms. These results will support future studies to reveal the powdery mildew resistance mechanisms. The two novel *er1* alleles and the developed co-dominant functional markers could be powerful tools for the breeding of pea cultivars resistance to *E. pisi*.

## 4. Materials and Methods

### 4.1. Plant Material and E. pisi Inoculum

Previously, many Chinese pea germplasms had been screened for *E. pisi* and some were found to be *E. pisi*-resistant [31,37,39]. In this study, the three *E. pisi*-resistant Chinese pea landraces, Suoshadabaiwan, Dabaiwandou, and Guiwan 1, respectively, from the Chongqing, Yunnan, and Guangxi provinces of China were conducted to reveal their *E. pisi*-resistant genes. The three *E. pisi*-susceptible Chinese pea cultivars, Bawan 6, Longwan 1, and Chengwan8, were used as susceptible controls or cross susceptible parents for genetic analysis [15,40]. The Chinese pea cultivar Xucai 1 containing *er1*-2 was used as *E. pisi*-resistant control [16].

The *E. pisi* isolate EPYN from Yunnan Province of China was used as the inoculum, which is highly virulent on pea [17,18]. The EPYN isolate was maintained through the continuous re-inoculation of healthy seedlings of Longwan 1 under controlled conditions. The inoculated plants were incubated in a growth chamber with controlled conditions [16].

### 4.2. Powdery Mildew Resistance Evaluation

Thirty-five seeds were planted from each of the three *E*. *pisi*-resistant germplasms (Suoshadabaiwan, Dabaiwandou, and Guiwan 1), three *E*. *pisi*-susceptible pea cultivars (Bawan 6, Longwan 1, and Chengwan 8), and from the resistant and susceptible controls (Bawan 6, Longwan 1 and Chengwan 8, and Xucai 1) [18]. The healthy seedlings were thinned to 30 per pot before the phenotypic evaluation. Three replications were planted. Seeded pots were placed in a greenhouse maintained at 18 to 26 °C. At the same time, the *E. pisi* inoculum was prepared by inoculating the 10-day-old seedlings of Longwan 1, which were incubated in a growth chamber at 20 ± 1 °C with a 12-h photoperiod. Two weeks later, all seedlings of the germplasms and controls were inoculated by gently shaking off the conidia of the Longwan 1 plants. Inoculated plants were incubated in a growth chamber at 20 ± 1 °C with a 12-h photoperiod. Ten days later, disease severity was rated based on a scale (0–4 scale) [17,18]. Plants with a score of 0 were considered *E*. *pisi*-immune, while those with scores of 1, 2 and 3, 4 were considered E. pisi-resistant and E. pisi-susceptible, respectively. For those identified as immune or resistant to *E*. *pisi*, repeated identification was performed.

### 4.3. Inheritance Analysis of Resistant Pea Cultivars

To reveal the inheritance controlled by *E. pisi* resistance genes in the three *E. pisi*-resistant Chinese pea landrace, Suoshadabaiwan, Dabaiwandou, and Guiwan 1, they were crossed with the *E. pisi*-susceptible cultivars Bawan 6, Longwan 1, and Chengwan 8, respectively, to generate genetic populations. The derived F_1_, F_2_, and F_2:3_ populations from three crosses (Bawan 6 × Suoshadabaiwan, Longwan 1 × Dabaiwandou, and Chengwan 8 × Guiwan 1), which were used for the powdery mildew resistance genetic analysis of Suoshadabaiwan, Dabaiwandou, and Guiwan 1. The six parents and the derived F_1_ and F_2_ populations were planted in a propagation greenhouse to generate F_2_ and F_2:3_ family seeds, respectively.

The plants of the F_1_ and F_2_ populations at the fourth or fifth leaf stage were inoculated with the *E. pisi* isolate EPYN using the detached leaf method [16,17,18]. After inoculation, the treated leaves were placed in a growth chamber at 20 °C with a 14-h photoperiod. The six parents were also inoculated as controls. Ten days after inoculation, disease severity was rated based on a scale of 0–4 as described above. Plants with scores of 0–2 and 3–4 were classified as resistant and susceptible, respectively [16,17,18]. Those plants identified as *E. pisi*-resistant were tested again to confirm their resistance.

Twenty-five seeds were selected randomly from each of the 102, 121, and 131 F_2:3_ families derived from Bawan 6 × Suoshadabaiwan, Longwan 1 × Dabaiwandou, and Chengwan 8 × Guiwan 1, respectively. These seeds were planted and cultivated together with their parents, following previously published protocols [25,26,27]. Disease severity was scored 10 days after inoculation using the 0–4 scale, as described above for the phenotypic identification of the pea germplasms. The F_2:3_ families with scores of 0–2 and 3–4 were classified as homozygous resistant and homozygous susceptible, respectively. Families with scores of 0–2 and 3–4 were considered segregated to *E. pisi* resistance. The families identified as homozygous-resistant or resistance segregated were subjected to repeated testing.

A chi-squared (χ^2^) analysis was used to evaluate the goodness-of-fit to Mendelian segregation ratio of the F_2_ and F_2:3_ phenotypes derived from Bawan 6 × Suoshadabaiwan, Longwan 1 × Dabaiwandou, and Chengwan 8 × Guiwan 1, respectively.

### 4.4. Genetic Mapping of the Resistance Gene

The Genomic DNA was isolated from the leaves of the F_2_ populations and of their parents using a slightly modified cetyltrimethylammonium bromide (CTAB) extraction method [42]. The DNA solution was diluted and stored at −20 °C until use.

Previous studies suggested that *E. pisi* resistance was controlled by *er1* in most of all pea germplasms except for lines SVP952 and JI 2480 [7,21]. Thus, this study was performed to map the *E. pisi*-resistance gene using the known *er1*-linked markers on the pea LG VI, including five simple sequence repeat (SSR) markers (PSMPSAD51, PSMPSA5, PSMPSAD60, i.e., AD60, PSMPSAA374e, and PSMPSAA369); a gene marker (Cytosine-5, DNA-methyltransferase (c5DNAmet)) [12,15,16,17,18,37,43]; and 10 molecular markers on the pea LG VI (AD160, AC74, AC10_1, AA224, AA200, AD159, AD59, AB71, AA335, and AB86). Firstly, these markers were used to screen for polymorphisms between the crossed parents, Bawan 6 × Suoshadabaiwan, Longwan 1 × Dabaiwandou, and Chengwan 8 × Guiwan 1 [44]. The parental polymorphic markers were then used to confirm the genotypes of each F_2_ plant for genetic linkage analysis. PCR amplification was conducted in a total volume of 20 µL [16,17,18]. PCR reactions were performed in a thermal cycler (Biometra, Göttingen, Germany). The PCR products were separated on 6%–8% polyacrylamide gels.

The segregation data of the polymorphic markers in the F_2_ populations were evaluated for goodness-of-fit to Mendelian segregation patterns with a chi-squared (χ^2^) test. Genetic linkage analyses were completed using MAPMAKER/EXP version 3.0b. A logarithm of odds (LOD) score > 3.0 and a distance < 50 cM were used as the thresholds to determine the linkage groups [45]. Genetic distances were determined using the Kosambi mapping function [46]. The genetic linkage map was constructed using the Microsoft Excel macro MapDraw [47].

### 4.5. RNA Extraction and PsMLO1 Sequence Analysis

The extraction of total RNA and synthesis of cDNA from Suoshadabaiwan, Dabaiwandou, and Guiwan 1 and controls were completed according to our previous studies [16,17,18].

To identify the resistance gene *er1* alleles, the full-length cDNAs of the *PsMLO1* homologs were amplified using the primers specific to *PsMLO1* [9]. The PCR cycling conditions were as follows: 95 °C for 5 min; then 35 cycles of denaturation at 94 °C for 30 s, annealing at 58 °C for 45 s, and extension at 72 °C for 1 min; and a final extension at 72 °C for 10 min. The purified amplicons were cloned with a pEasy-T5 vector (TransGen Biotech, Beijing, China). The sequencing reactions of 10 clones per parental cultivars and controls were performed by the Shanghai Shenggong Biological Engineering Co., Ltd. (Shanghai, China). The resulting sequences were aligned with the wild-type *PsMLO1* of pea (NCBI accession number: FJ463618.1) using DNAMAN v6.0 (Lynnon Biosoft, Vaudreuil, QC, Canada).

### 4.6. Development of Functional Markers for the Novel er1 Alleles

Primers flanking the mutation site were designed based on the *PsMLO1* gene sequence (GenBank accession number KC466597), using Primer Premier v5.0, to develop a functional marker specific to allele *er1*-13 and *er1*-14 (Table 2).

The Kompetitive allele-specific PCR (KASPar) markers (KASPar-*er1*-13 and KASPar-*er1*-14) specific to the two novel *er1* alleles were developed based on allele *er1*-13 SNPs (1-bp deletion) and *er1*-14 SNPs (1-bp deletion) in *PsMLO1*. The forward primers and the common reverse primers specific to *er1*-13 and *er1*-14 were designed for KASPar markers by LGC KBioscience (KBioscience, Hoddesdon, UK), respectively. In brief, two KASPar markers (KASPar-*er1*-13 and KASPar-*er1*-14) were used to detect parental polymorphisms (Longwan 1 × Dabaiwandou, and Chengwan 8 × Guiwan 1), and then used to analyze the genotypes of each F_2_ offspring (Longwan 1 × Dabaiwandou: 121 F_2_ individuals; Chengwan 8 × Guiwan: 131 F_2_ individuals).

KASPar markers were amplified with a Douglas Scientific Array Tape Platform (China Golden Marker Biotech Co., Ltd., (Beijing, China)) in a 0.8 µL Array Tape reaction volume with 10 ng dry DNA, 0.8 µL 2 × KASP master mix, and 0.011 µL primer mix (KBioscience, Hoddesdon, UK). A Nexar Liquid handling instrument was used to add the PCR solution to the Array Tape (Douglas Scientific). PCRs were performed on a Soellex PCR Thermal Cycler with the following conditions: initial denaturation at 94 °C for 15 min; followed by 10 cycles of denaturation at 94 °C for 20 s and 65 °C for 60 s at an annealing temperature that decreased by 0.8 °C per cycle; and then 26 cycles of denaturation at 94 °C for 20 s and 57 °C for 60 s; and a final cooling to 4 °C. A fluorescent end-point reading was completed with the Araya fluorescence detection system (part of the Douglas Scientific Array Tape Platform). Genotypes and clusters were visualized with Kraken (http://ccb.jhu.edu/software/kraken/MANUAL.html (accessed on 5 August 2022)).

### 4.7. Validation and Application of Functional Markers

To test the efficacy of the novel functional markers specific to *er1*-13 (KASPar-*er1-13*) and *er1*-14 (KASPar-*er1*-14), 56 pea germplasms with known phenotypic resistance to *E. pisi* isolate EPYN and carrying known *er1* alleles, and six parents were tested for whether they carried the *er1* alleles *er1*-13 or *er1*-14 (Appendix A) [9,15,17,18,32,40]. The six parents (Suoshadabaiwan, Dabaiwandou and Guiwan 1, Bawan 6, Longwan 1, and Chengwan 8) were used as contrasting controls.

DNA was extracted from the 56 selected pea germplasms (resistant cultivars with known *er1* alleles) and the six parents using the CTAB method [42]. The PCR amplification of the KASPar markers were performed as described above (in the section “Development of Functional Markers for the novel *er1* alleles”).

## Figures and Tables

**Figure 1 ijms-23-12016-f001:**
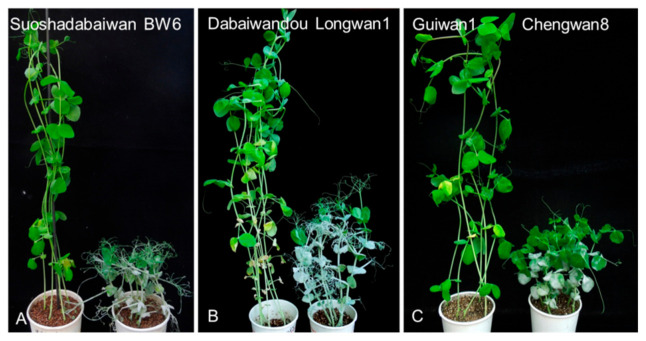
Phenotypic evaluation of the three *Erysiphe pisi*-resistant pea cultivars Suoshadabaiwan, Dabaiwandou, and Guiwan 1, as well as the three *E. pisi*-susceptible cultivars Bawan 6, Longwan 1, and Chengwan 8, after inoculation with *E. pisi* isolate EPYN. (**A**) Suoshadabaiwan and *E. pisi*-susceptible cultivar Bawan 6 (BW6). (**B**) Dabaiwandou and *E. pisi*-susceptible cultivar Longwan 1. (**C**) Guiwan 1 and *E. pisi*-susceptible cultivar Chengwan 8.

**Figure 2 ijms-23-12016-f002:**
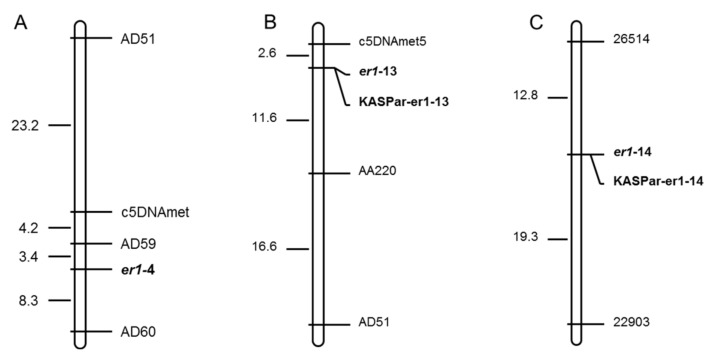
Genetic linkage maps constructed using the *er1*-linked markers and the functional markers for *er1*-13 and *er1*-14, based on the F_2_ populations derived from (**A**) Bawan 6 × Suoshadabaiwan, (**B**) Longwan 1 × Dabaiwandou, and (**C**) Chengwan 8 × Guiwan 1. Map distances and loci order were determined with MAPMAKER v3.0. Estimated genetic distances between loci are shown to the left of the maps in centiMorgans (cM).

**Figure 3 ijms-23-12016-f003:**
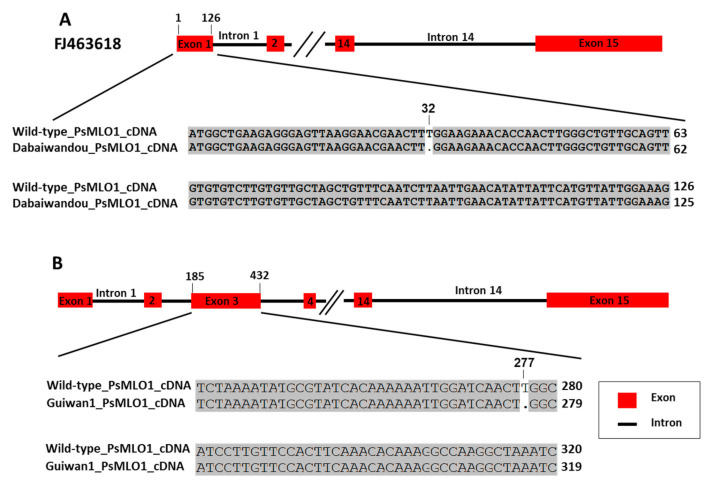
*PsMLO1* cDNA sequence comparisons of those from the powdery mildew-resistant pea landrace Dabaiwandou and Guiwan 1 with the wild-type pea cultivar Sprinter (GenBank accession number: FJ463618.1). (**A**) There is a single base deletion (T) in the *PsMLO1* cDNA of Dabaiwandou at positions 32 of exon 1. (**B**) There is a single base deletion (T) in the *PsMLO1* cDNA sequence of Guiwan 1 at position277 in exon 3. The figure shows the difference of nucleotide sequence from Dabaiwandou, Guiwan 1, and the wild-type pea cultivar Sprinter. The two mutation sites are indicated in the respective cDNA sequences.

**Figure 4 ijms-23-12016-f004:**
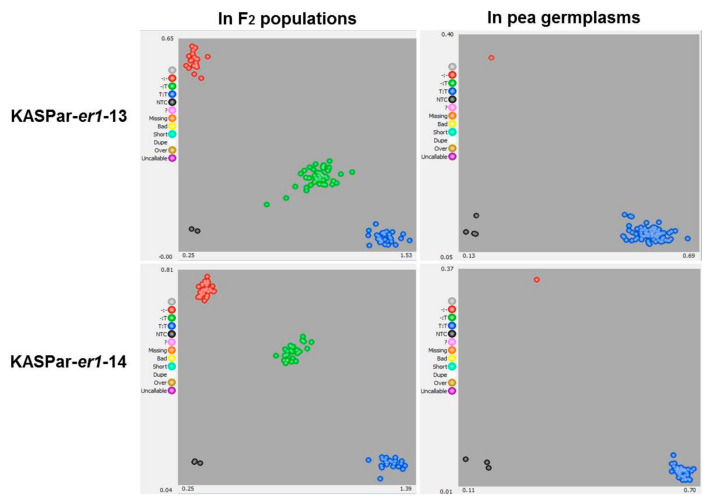
KASPar genotyping detection with markers KASPar-er1-13 and KASPar-er1-14 in the F_2_ populations derived from Longwan 1 × Dabaiwandou and Chengwan 8 × Guiwan 1, as well as in other pea germplasms. Red dots indicate er1-13/er1-14 homozygous *Erysiphe pisi*-resistant lines/germplasms, blue dots indicate Er1-13/Er1-14 homozygous *E. pisi*-susceptible lines/germplasms, and green dots indicate er1-13/Er1-13 or er1-14/Er1-14 heterozygotes. Grey dots are blank samples used as controls. In our KASPar assay, the co-dominant markers KASPar-er1-13 and KASPar-er1-14 correctly categorized all F_2_ individuals into three clusters corresponding to homozygous resistant (red dots), homozygous susceptible (blue dots), and heterozygous (green dots) lines, and categorized tested pea germplasms into two clusters corresponding to homozygous resistant (red dots) and homozygous susceptible (blue dots).

**Table 1 ijms-23-12016-t001:** Segregation patterns of pea resistance to powdery mildew in the F_1_, F_2_, and F_2:3_ populations derived from three crosses, Bawan 6 × Suoshadabaiwan, Longwan 1 × Dabaiwandou, Chengwan 8 × Guiwan 1.

Parents and the Cross	Generation	Amount	No. of Plant or Families			Expected Ratio and Goodness of Fit		
			R	Rs	S	R:Rs:S	χ^2^	P
Bawan 6	P1	30	-	-	30	-		
Suoshadabaiwan	P2	30	30	-	-	-		
Bawan 6 × Suoshadabaiwan	F_1_	5	-	-	5	-		
	F_2_	102	26	-	76	1:3	0.02	0.88
	F_2:3_	102	26	51	25	1:2:1	0.03	0.99
Longwan 1	P_1_	30	-	-	30	-		
Dabaiwandou	P_2_	30	30	-	-	-		
	F_1_	6	-	-	6	-		
Longwan 1 × Dabaiwandou	F_2_	121	29	-	92	1:3	0.07	0.79
	F_2:3_	121	29	56	36	1:2:1	1.41	0.49
Chengwan 8	P_1_	30	-	-	30	-		
Guiwan 1	P_2_	30	30	-	-	-		
	F_1_	8	-	-	8	-		
Chengwan 8 × Guiwan 1	F_2_	131	36	-	95	1:3	0.43	0.51
	F_2:3_	131	36	61	34	1:2:1	0.67	0.71

“R”, “Rs”, and “S” stand for resistant, segregating, and susceptible, respectively.

**Table 2 ijms-23-12016-t002:** Sequence information for the indel and Kompetitive allele-specific PCR (KASPar) markers specific to *er1*-13 and for the KASPar marker specific to *er1*-14.

Markers	Primers	Sequence Information (5′–3′)	Annealing Tm
KASPar-*er1-13*	Forward-T	GAAGAGGGAGTTAAGGAACGAACTTT	65–57 °C touchdown
	Forward	AAGAGGGAGTTAAGGAACGAACTTG	
	Common reverse	TGCAACAGCCCAAGTTGGTGTTTCT	
KASPar-*er1-14*	Forward-T	ATATGCGTATCACAAAAAATTGGATCAACTT	65–57 °C touchdown
	Forward	GCGTATCACAAAAAATTGGATCAACTG	
	Common reverse	CCTTGGCCTTTGTGTTTGAAGTGGAA	

## Data Availability

The data presented in this study are available on request from the corresponding author.

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
