# Peer review of "Molecular Characterizations of the er1 Alleles Conferring Resistance to Erysiphe pisi in Three Chinese Pea (Pisum sativum L.) Landraces"

_ijms, 2022, doi:10.3390/ijms231912016_

Round 1

Reviewer 1 Report

Please see the comments.

Author Response

Reviewer1: The work by Sun et al. advances our understating of disease tolerance in Pea. It explores the novel aspect for disease resistance in Pea. The report is well written and condensed, as well as technically appropriate. However, before being able to recommend acceptance, I invite authors to address the following minor amendments. Abstract should be reduced to 250-300 words. Response: Thanks for your good suggestion. We have deleted the unnecessary words or sentences in abstract section and reduced to less than 300 words. In line 47, please mention disease name. Response: In line 47, the content is : “crop, which frequently suffers from abiotic and biotic stresses during the whole growth”, I think it is unnecessary to add the disease name in this line. The following content includes the disease name in the manuscript. In line 53-58, author mention er3 a dominant gene involved in disease resistance, why author did not work on er3 genes for research? Why only recessive genes selected for study? Response: Because Er3 is a dominant gene and from wild pea Pisum fulvum. It is a pity that we have no the material used to study. Moreover, recessive gene er1 controls durable and broad-spectrum resistance to Erysiphe pisi. Author hypothesis the crosses between resistance and susceptible plants could brought resistance population in F1, F2, or F3 population. Author must read the article https://doi.org/10.1080/15440478.2020.1838996. Response: Thanks for providing the article. We read the article you provided(Genetic Variation Studies of Ionic and within Boll Yield Components in Cotton (Gossypium Hirsutum L.). The article is about abiotic tress (salt) and the tolerance is controlled by multiple genes under salt. We think it is a good study, but not suitable for citing in our study. (Marked in yellow color). Although the paper provides good evidence into the disease tolerance in some F2 families of Pea, a major question that authors should prospect in their discussion is how Pea improvement for disease tolerance may unlock and effectively utilize hidden variation from the wild genepools (please refer to and include Agronomy 10.3390/agronomy12061310 ). Response: Thanks for providing the article (Biochemical and Associated Agronomic Traits in Gossypium hirsutum L. under High Temperature Stress). We read the article you provided. The article is about abiotic tress (High Temperature Stress). Heat tolerance is a physiologically and genetically complex trait regulated by multiple genes. We think it is a good study, but not suitable for citing in our study. Inter-specific crossing may offer an avenue in this regard that authors should acknowledge (please refer to and include Agronomy 10.3390/agronomy12061310). Response: Thanks for your good comment. We added the contents about discovery or effectively utilize resistance genes from the wild germplasms in the discussion section.

Reviewer 2 Report

In this study, authors identified the resistance gene for E. pisi in the three Landraces by genetic mapping and cloning. er1-4 was contained in Suoshadabaiwan. Whereas, two novel er1 alleles, er1-13 and er1-14, were identified in Dabaiwandou and Guiwan 1, respectively. The functional markers for er1-13 and er1-14 were developed and validated in populations and pea germplasms. These results would provide helpful information of E. pisi resistance in Chinese pea landraces and molecular markers for pea breeding. However, the paper needs improvement before acceptance for publication.

My detailed comments are as follows:

1. Authors should provide more common markers in Figure 2 and revise the genetic map in the same orientation to suggest that three resistance genes in Chinese pea are allelic to er1 locus.

2. Furthermore, authors should do cross and conducted allelic test between the known allele of er1 and two novel alleles, er1-13 and er1-14.

Author Response

In this study, authors identified the resistance gene for E. pisi in the three Landraces by genetic mapping and cloning. er1-4 was contained in Suoshadabaiwan. Whereas, two novel er1 alleles, er1-13 and er1-14, were identified in Dabaiwandou and Guiwan 1, respectively. The functional markers for er1-13 and er1-14 were developed and validated in populations and pea germplasms. These results would provide helpful information of E. pisi resistance in Chinese pea landraces and molecular markers for pea breeding. However, the paper needs improvement before acceptance for publication. My detailed comments are as follows: 1. Authors should provide more common markers in Figure 2 and revise the genetic map in the same orientation to suggest that three resistance genes in Chinese pea are allelic to er1 locus. Response: Thanks for your comment. In this study, we used the different parents to generate the genetic populations for genetic linkage analysis, the polymorphic markers are different between respective parents. It is difficult to make the genetic maps in the same orientation. Moreover, homologous PsMLO1 gene identification and development of functional markers based on homologous PsMLO1 gene indicate the resistance genes in three landraces are allelic to er1 locus. 2. Furthermore, authors should do cross and conducted allelic test between the known allele of er1 and two novel alleles, er1-13 and er1-14. Response: Thanks for your comment. This study identified the three er1 alleles, er1-4, er1-13 and er1-14, they are all from the mutations of PsMLO1 gene. Thus, they are alleles of er1. We directly clone PsMLO1 gene and verified them by functional markers developed. We think allelic test is not necessary.

Round 2

Reviewer 2 Report

The authors improved the presentationand gave response to my comments. I suggest acceptance of the manuscript.

Author Response

Thanks for your conmments and acceptance of the manuscript. English language and style were checked and made minor revisions.